# Pharmacological Treatment of Alcohol Cravings

**DOI:** 10.3390/brainsci13081206

**Published:** 2023-08-15

**Authors:** Matheus Cheibub David Marin, Maria Olívia Pozzolo Pedro, Giuliana Perrotte, Anderson S. Martins-da-Silva, Dangela L. S. Lassi, Israel Kanaan Blaas, Fernando Ikeda Castaldelli, Maria Beatriz Brisola dos Santos, Guilherme Trevizan Kortas, Marcela Waisman Campos, Julio Torales, Antonio Ventriglio, Cintia de Azevedo-Marques Périco, André B. Negrão, Kae Leopoldo, Arthur Guerra de Andrade, André Malbergier, João Maurício Castaldelli-Maia

**Affiliations:** 1Perdizes Institute (IPer), Clinics Hospital of the Medical School (HCFMUSP), University of São Paulo, São Paulo 05021-001, Brazil; dr.matheus@gmail.com (M.C.D.M.); dangela.lassi@hc.fm.usp.br (D.L.S.L.); israel_blaas@hotmail.com (I.K.B.); gtkortas@gmail.com (G.T.K.); abnegrao@usp.br (A.B.N.); aandrade@usp.br (A.G.d.A.); andre.malbergier@hc.fm.usp.br (A.M.); 2Hospital Sírio-Libanês, São Paulo 01308-050, Brazil; maria.oliviapozzolo@gmail.com (M.O.P.P.); biabrisola@gmail.com (M.B.B.d.S.); 3Department of Psychiatry, Medical School, University of São Paulo, São Paulo 05403-903, Brazil; andersonsmsilva@gmail.com (A.S.M.-d.-S.); kae.leopoldo@usp.br (K.L.); 4Department of Neuroscience, Medical School, FMABC University Center, Santo André 09060-870, Brazil; giu.perrotte@gmail.com (G.P.); cazevedomarques@hotmail.com (C.d.A.-M.P.); 5Medical School Catanduva (FAMECA), Padre Albino University Center, Catanduva 15809-144, Brazil; fernando.ikeda@hotmail.com; 6Department of Cognitive Neurology, Neuropsychiatry, and Neuropsychology, Fleni, Buenos Aires C1428AQK, Argentina; mwconsultorio@hotmail.com; 7Department of Medical Psychology, School of Medical Sciences, Universidad Nacional de Asunción, San Lorenzo 111454, Paraguay; juliotorales@gmail.com; 8Regional Institute of Health Research, Universidad Nacional de Caaguazú, Coronel Oviedo 050106, Paraguay; 9School of Health Sciences, Universidad Sudamericana, Pedro Juan Caballero 130112, Paraguay; 10Department of Experimental Medicine, Medical School, University of Foggia, 71100 Foggia, Italy; a.ventriglio@libero.it

**Keywords:** craving, alcohol, randomi* controlled blind*

## Abstract

(1) Background: The treatment of substance addiction is challenging and has persisted for decades, with only a few therapeutic options. Although there are some recommendations for specific treatments for Alcohol Use Disorder (AUD), there is no specific medication used to treat alcohol cravings, which could benefit millions of patients that are suffering from alcoholism. Cravings, or the urge to use drugs, refer to the desire to experience the effects of a previously experienced psychoactive substance. (2) Methods: We included original studies of alcohol abuse or dependence extracted from a controlled, blind, pharmacological treatment study which presented measures and outcomes related to alcohol cravings. (3) Results: Specific drugs used for the treatment of alcoholism, such as Naltrexone and Acamprosate, have had the best results in relieving craving symptoms, as well as promoting abstinence. Baclofen and anticonvulsants such as Gabapentin and Topiramate have shown good results in promoting abstinence and the cessation of cravings. (4) Conclusions: Specific drugs used for the treatment of alcoholism to obtain the best results can be considered the gold standard for promoting abstinence and relieving cravings. Anticonvulsants and Baclofen also had good results, with these medications being considered as second-line ones. Varenicline is an option for alcohol dependents who also concomitantly ingest tobacco.

## 1. Introduction

Alcohol Use Disorder (AUD) is a worldwide disease with strong impacts not only on people’s health, but also on the economy and society. The result is a greater number of sick patients, with consequent expenses due to hospitalizations and the involvement of a portion of the economically active population. According to the WHO (World Health Organization) surveys, worldwide, three million deaths a year result from the harmful use of alcohol, representing 5.3% of all deaths. The harmful use of alcohol is a causal factor for over 200 illnesses and injuries. Overall, 5.1% of the global burden of diseases and injuries is attributable to alcohol consumption, as calculated in terms of Disability-Adjusted Life Years Lost (DALY). Alcohol consumption causes death and disability relatively early in life. In the 20–39 age group, approximately 13.5% of the total deaths are attributable to alcohol. Unfortunately, treatment options aimed at abstinence are scarce and have poor results [1].

Despite this, there are numerous outcomes beyond abstinence that can improve an individual’s quality of life, given the immense difficulty for addicted patients to remain abstinent. An alternative to reducing their alcohol consumption or even achieving abstinence would be offered to treat the cravings.

Cravings were recognized for the first time by Jellinek and colleagues in 1955 as a central component of alcohol dependence syndrome [2]. In the most specific sense, cravings refers to the subjective longing or desire to use drugs. Within this perspective, the subjective experience of cravings is considered to be the primary aspect of and driving force behind drug use [2].

The contemporary theoretical framework regarding the phenomenon of cravings incorporates behavioral theories of classical conditioning, cognitive and motivational mechanisms, as well as anatomofunctional and neurobiochemical factors [3]. According to the Diagnostic and Statistical Manual of Mental Disorders (DSM) 5 cravings are one of the diagnostic criteria for alcohol dependence syndrome and is defined as “the desire to experience the effects of a previously experienced psychoactive substance” [4].

Therefore, in treating the impending and often uncontrollable urge to use drugs, the patient will use the substance less often, causing less harm. Specific medications for alcoholism continue to be the first options for both curbing cravings and maintaining abstinence. Opioid antagonists, especially Naltrexone, are drugs with the best responses [5,6,7,8,9,10].

There are several medications studies used for the treatment of AUD, including anticonvulsants, psychotropic drugs, and anti-smoking drugs. Other authors have presented some important results, and they can be considered as second-line medications or in refractory cases [11,12,13,14,15,16,17,18,19,20,21].

Unfortunately, there are no summaries or guidelines for the effective pharmacological treatment of cravings. Clinical trials are usually designed with an abstinence endpoint, but many of them collect data on secondary endpoints, such as cravings. Perhaps this explains the reason for there being only a few studies on this pathology, which requires attention and specifies treatment.

The objective of this review is to correlate the possible treatments that cause a response to cravings, as well as to promote abstinence.

## 2. Materials and Methods

### 2.1. Eligibility

Original clinical trials testing a pharmacological treatment for cravings in alcohol abusers or dependents were included in this review. We excluded non-original studies (i.e., reviews, meta-analysis, case reports, discussion articles, and study protocols) in this review. We also excluded those which were not randomized, controlled, or double-blinded; studies in languages other than English; those that used animals; those that did not include alcohol abusers or dependents; those analyzing the non-pharmacological treatment of alcohol cravings; and studies evaluating other alcohol measures than cravings.

### 2.2. Information Sources

Databases from Pubmed of the US National Library of Medicine and PsycINFO were used to identify relevant studies.

### 2.3. Search Strategy

The keywords used in the search were ‘craving’ and ‘alcohol’ and ‘randomi* controlled Blind*’. ‘Clinical trial’ and ‘humans’ filters were used in the Pubmed and PsycINFO databases.

### 2.4. Study Selection

To select articles for this review, the first and the last authors read the abstracts of all the studies found in the search. Duplicated articles and studies not related to alcohol were excluded. In the last step, the first author read the remaining studies. Inclusion and exclusion criteria were applied. The present review followed the PRISMA Statement for the transparent reporting of systematic reviews and meta-analyses [22] (Appendix A).

### 2.5. Data Collection Process

The first and last authors independently included studies. The quality assessment was performed using the Study Quality Guide from the Cochrane Consumers and Communication Review Group (2013), considering six domains of risk of bias: (1) random sequence generation; (2) allocation concealment; (3) the blinding of participants and personnel; (4) the blinding of the outcome assessment; (5) an incomplete data outcome; and (6) a selective report. In cases of disagreement between the first and the last authors on the information presented in the review, the second one decided on the best way to present the data.

### 2.6. Data Items

In this review, the authors searched for information on the following variables: the title of the study; author(s); year of publication; sample size; mean age; sample characteristics; setting; search for treatment; design of the study; craving measures and other relevant measures; intervention; outcomes; and drop-out rates.

### 2.7. Inclusion and Exclusion Criteria

We selected randomized the articles on double-blind, placebo-controlled clinical trials on patients with alcohol disorder who are dependent. Animal subjects, the use of other drugs, the presence of psychiatric comorbidities, and studies that did not assess cravings were excluded.

### 2.8. Selection Articles

A total of 442 articles were identified. First, we read the title of the article, the publication of the journal, and their summaries, excluding repeated studies, studies that did not meet the criteria, and those did not address the relevant issue. A total of 59 articles were excluded. For a second time, we read the remaining 383 studies. Among these, 10 articles were incompleted, 22 did not analyze cravings or were not clinical trials, 1 included animal subjects; 5 were not related to Alcohol Use Disorder; 14 studies were single-blind trials; 11 included other drugs than alcohol; 10 were non-placebo-controlled articles; and 8 involved psychiatric comorbidities. Due to the large volume, we divided them into 3 groups: 60 randomized, double-blind, placebo-controlled articles on alcohol dependents, 26 articles were on cue-induced cravings, and 9 articles on heavy drinkers or subjects who misuse alcohol. In this review, we only cover double-blind placebo-controlled studies, excluding articles that did not study alcohol dependents and experimental/cue-induced-cravings-related articles (Appendix B) [23].

### 2.9. Qualitative Analysis of Studies

The selected articles were qualitatively analyzed through the evaluation of 13 items following the Checklist for Randomized Controlled Trials (Critical Appraisal tools for use in Joanna Briggs Institute Systematic Reviews). Thirteen questions were applied to each article, with the following answers: Yes (with a value of 1 point), No (0), Unclear (0.5), and Not Applicable (1) (Appendix A).

A cut-off point of 10 was considered as the limit. Only one article had obtained a score below 10, and one article had a limit score of 10. All the other articles had a score above 10, with the average score of all studies being 12.7. These results demonstrate that the articles analyzed in this review are relevant and have only a few biases.

## 3. Results

There was a wide range of focuses among the selected studies. The studies use a population sample ranging from 11 and up to 493 participants, with an average *n* = 109 participants. The majority were outpatients, male, aged between 18 and 70 years, with an average age in fourth decade of life. All the participants had the diagnosis of alcohol dependence, following the criteria of the DSM-III, DSM-IV, and International Classification of Diseases (ICD-10). The oldest study found was in 1983, and the most recent one was conducted in 2020, with most of the studies coming from North America. Most of the studies, 47.6% (48), are from the US, 28.5% are European, 12.7% are from Asia and Russia, and 4.76% are Australian. Studies from Brazil and Canada account for the remaining 4.76%.

Most studies used the Obsessive Compulsive Drinking Scale (OCDS) as a measure (31.7%), followed by the Visual Analogue Scale (VAS) (28.5%), and then the Pennsylvania Alcohol Craving Scale (PACS) (19%). There were other scales that were used to a lesser extent, such as the Desire for Alcohol Questionnaire (Short DAQ), Clinical Institute Withdrawal Assessment for Alcohol Scale (CIWA-Ar), Tiffany Craving Questionnaire, Alcohol Craving Scale–Short Form, and self-reported craving scores.

The studies lasted for 3 days, and some lasted for up to 6 months, with the vast majority having 12 weeks of follow-up (41.2%) and a mean of 8.47 weeks per study. About dropouts, the average rate was 27.5%. Six studies had no dropouts, and three did not report the dropouts. The study with the highest number of dropouts had a rate of 85%.

The goal of this review is to assess which treatments obtained better results regarding cravings and to associate which ones contributed to a longer period of abstinence. By analyzing the studies qualitatively, 33 different drugs were tested, obtaining positive results concerning the control of cravings via the use of 15 drugs. In terms of abstinence, in 60 articles, 16 showed a longer period of abstinence compared to that of the placebo group, 21 studies showed no difference between the control and placebo groups, and in 13 articles, abstinence was not evaluated, only cravings.

### 3.1. Specific Drugs for the Treatment of Alcoholism 

Specific medications used for the treatment of alcoholism comprised the vast majority of interventions used in the studies (about 18 studies) (Table 1). Among them were Naltrexone (eleven), Acamprosate (four), Naltrexone and Acamprosate (two), and Nalmefene (one).

Naltrexone, a pharmacological agent acting as an antagonist on the opioid receptors, has demonstrated a notable efficacy in reducing craving when administered as a monotherapy, as evidenced by the more positive outcomes in six out of seven studies compared to those of the placebo groups [5,6,7,29,32,34,38]. It is important to consider the pharmacokinetic properties of Naltrexone, including its half-life of 3.9–10.3 h and prolonged terminal elimination-phase half-life of 96 h [39]. Notably, the major metabolite of Naltrexone, 6-beta-naltrexol, exhibits higher plasma concentrations than the parent drug does. Understanding these pharmacokinetic characteristics provides insights into the sustained effects and potential mechanisms of action of Naltrexone in cravings reduction. In five studies associated with other interventions, there was a favorable result for craving cessation in only one when associated with Ondansetron [24]. In the other studies: PUFAS (Polyunsaturated Fatty Acids) [28], Cognitive Behavioral Therapy (CBT) [25], CBT and Supportive Therapy (ST) [27], and Acamprosate [33] were combined, but they were not superior to the placebo treatment applied to reduce the patients’ cravings.

Acamprosate, a structural analogue of gamma-aminobutyric acid (GABA), is believed to exert effects on alcohol intake by influencing the calcium channels and modulating transmission along GABA and decreasing the level of glutamate activity in NMDA receptors. These actions are hypothesized to lead to the reduced positive reinforcement of alcohol consumption and attenuated cravings during withdrawal [8,36]. Despite these proposed mechanisms, the precise mode of action of Acamprosate remain unknown. Acamprosate was focused on in six articles, where it reduced the patients’ cravings more than the placebo in two articles [30,31,36,37]. Nalmefene, an antagonist at the mu- and delta-opioid receptors [40], exhibited a non-significant reduction in cravings compared to that of a placebo in one study [26]. Nalmefene shares a similar chemical structure to Naltrexone, but it may possess unique properties that allow tighter binding to the opioid receptors [41].

As for abstinence, opioid receptor antagonists such as Naltrexone and Nalmefene were analyzed in 12 studies. In 5 out of 11 studies, Naltrexone was superior to the placebo [25,27,33,38,42]. Nalmefene did not result in a better response than that of the placebo in the only study included. It is worth mentioning that for Naltrexone, despite not having presented a positive response for the reduction of cravings when associated with CBT or ST, there was a positive response in terms of maintaining abstinence [25,27]. Acamprosate was successful in maintaining abstinence in two out of four articles [31,33].

In two studies, Naltrexone was compared to Acamprosate and a placebo. In one article, Naltrexone was found to be superior at reducing cravings, while Acamprosate did not differ from the placebo [35]. In the other paper, a comparison was made between Naltrexone, Acamprosate, placebo, and a combination of Naltrexone and Acamprosate, with the aim to correlate it to an increasing leptin plasma concentration. There was no association between the baseline plasma leptin concentration and cravings. However, in the group with the combined medication, decreasing leptin concentrations correlated significantly with the duration of abstinence [33].

### 3.2. Anticonvulsants 

Another important group of drugs that is widely used in psychiatry is anticonvulsants. These drugs have three mechanisms of action: (a) the potentiation of GABA action; (b) the inhibition of sodium channel function; and (c) the inhibition of calcium channel function. By promoting central nervous system depression, these drugs act on one of the receptors that alcohol acts on, simulating its effect and attenuating the desire or cravings. Ten articles which used anticonvulsants were found: Gabapentin (three), Topiramate (three), Gamma-hydroxybutyric acid (GHB) (one), Pregabalin (one), Oxcarbazepine (one), and Levetiracetam (one) (Table 2).

Gabapentin is a synthetic amino acid that serves as a structural analog of GABA. It acts by binding to the alpha-2-delta type 1 subunit of voltage-gated calcium channels. Through this mechanism, Gabapentin selectively inhibits the influx of calcium ions (Ca^2+^) via these channels. By reducing the post-synaptic excitability and modulating the release of excitatory neurotransmitters, Gabapentin can decrease the level of neuronal excitability. Moreover, Gabapentin is believed to increase the concentration of GABA in the brain, possibly by blocking excitatory neurotransmission [52,53,54,55]. About Gabapentin, two articles were found to have evaluated cravings, showing a benefit superior to that of the placebo, and in one of them, there was the maintenance of abstinence, while the other article did not evaluate abstinence [14,47]. An important fact is that Gabapentin enacarbyl extended release was used in one study, which had negative results in an attempt to reduce the cravings and promote abstinence [13].

Topiramate was analyzed in three articles, demonstrating a favorable response in two to the cessation of cravings and in maintaining abstinence [16,51]. Topiramate, initially developed as an anti-diabetic drug and later used as an anticonvulsant, shares structural similarities with acetazolamide. It acts as a positive allosteric modulator at the GABA A receptors, promoting the increased influx of chloride ions into neurons and enhancing the overall GABA-mediated inhibition. These effects are likely mediated through non-benzodiazepine binding sites on GABA A receptors [9,56].

GHB showed positive results for abstinence and craving reduction in the only article that was found. GHB is a short-chain fatty acid that occurs naturally in the central nervous system (CNS) in humans and is utilized for both recreational and therapeutic purposes [57,58].

Pregabalin, an α2δ voltage-gated calcium channel subunit ligand, and Levetiracetam, a second-generation antiepileptic drug, which exhibit their effects through various mechanisms that are not fully understood [52,53,59,60], showed no benefit over the placebo to reduce the cravings [16,51]. About abstinence, this was analyzed in a study on Pregabalin, which was favorable in this article [3,46].

A drug that is not an anticonvulsant, but has a similar action, is Baclofen. This medication is responsible for the treatment of muscle spasticity with a highly effective medullary action, and it depresses the transmission of the monosynaptic and polysynaptic reflexes through the stimulation of GABA B receptors. This stimulation, in turn, inhibits the release of excitatory amino acids, glutamate and aspartate. In six articles, Baclofen showed a significant reduction in cravings compared to the placebo in three articles [43,45,49]. About abstinence, there was a positive response in three out of six articles [43,48,50].

Oxcarbazepine is an anticonvulsant that blocks voltage-gated sodium channels, resulting in the stabilization of hyperexcited neural membranes, the inhibition of repetitive neuronal discharge, and the decreased propagation of synaptic impulses. Like other anticonvulsants, it could reduce cravings by reducing the level of neuronal excitability. However, in only one study, Oxcarbazepine did not differ from the placebo in reducing cravings [44].

### 3.3. Varenicline 

As an important drug and the gold standard in the treatment of smoking, Varenicline binds with a high affinity and selectivity to the α4β2 neuronal nicotinic acetylcholine receptors, where it acts as a partial agonist. Since many alcoholic patients also smoke, which is an associated comorbidity, the hypothesis of a possible treatment for smokers who are also alcohol-dependent was raised. Varenicline was effective in reducing alcohol cravings in alcohol-dependent smokers in the two selected studies. Among them, only one study evaluated the possible benefit of abstinence, with a favorable result when it was compared with that of the placebo [12,61] (Table 3).

Mecamylamine, a nicotine antagonist, is a ganglionic cholinergic blocker, which was originally marketed for blood pressure lowering, and its use in smoking cessation blocks the rewarding effects of nicotine, and therefore, reduces the urge to smoke. The studied paper had a different result compared to that of Varenicline, showing a larger non-significant reduction in cravings than that of the placebo and no effect on promoting abstinence among smokers with alcohol dependence [62] (Table 3).

### 3.4. Other Psychotropic Drugs 

Memantine, Fluoxetine, Olanzapine, Samidorphan, Amisulpride, Buspirone, Ritanserin, Citalopram, and Quetiapine were all used in one individualized study for each (Table 4). Memantine acts on the glutamatergic system by blocking the N-methyl-d-aspartate-type glutamate receptors (NMDARs) [63]. Fluoxetine and Citalopram are selective serotonin reuptake inhibitors (SSRIs) and antidepressants used for the treatment of people with co-occurring depression and alcohol dependence [64,65]. Samidorphan is a new chemical entity that, in vivo, has been demonstrated to function as a μ-opioid antagonist [66]. Olanzapine, Amisulpride, and Quetipine are atypical antipsychotic drugs used in dual disorders [21,67]. Buspirone acts as a 5-HT1A receptor agonist for anxiety symptoms [68], and ritanserin is a serotonin antagonist drug described as an anxiolytic, antidepressant, antiparkinsonian, and antihypertensive drug. However, ritanserin has never been marketed for medical use due to safety issues, but it is currently used in scientific research [17,69].

The ones that promoted the increased reduction of cravings compared to that of the placebo were Fluoxetine, Samidorphan, and Quetiapine [66,72,73], and Quetiapine was favorable in maintaining abstinence [66].

Memantine had inferior results in maintaining abstinence and a larger non-significant reduction in cravings than that of the placebo. Additionally, it resulted in numerous side effects, which led to a drug dose reduction or discontinuation [70]. Olanzapine did not differ from the placebo in reducing cravings or maintaining abstinence [71].

### 3.5. Other Drugs 

Numerous drugs which are not specifically used for the treatment of AUD are used in psychiatry with a certain frequency due to the scarcity of therapeutic options.

In view of this, eight different drugs used widely in medicine, but are not previously indicated for substance use disorder (SUD) were tested in different 10 studies: Bromocriptine (2), Prazosin (2), N_2_O (1), Acetyl-L-carnitine (1), Atenolol (1), Dextromethorphan (1), Doxazosin (1), Intranasal Oxytocin (1), Ondansetron (1), Kudzu Root Extract, (1) and Enalapril (1) (Table 5).

Bromocriptine, a derivative of ergot alkaloid, exhibits dual properties as both a dopamine agonist and antagonist. It has been extensively investigated in various psychiatric disorders, covering a wide range of conditions [85]. Prazosin, which functions as an alpha1-adrenergic antagonist, effectively mitigates adrenergic hyperactivity and excessive drinking associated with alcohol withdrawal in laboratory animals. Moreover, it has demonstrated the ability to reduce the rate of alcohol seeking triggered by stress and lower the risk of relapse in individuals with AUD [10,20,86]. Carnitine plays a crucial role in facilitating the transportation of long-chain fatty acids (FAs) into mitochondria for β-oxidation. This process is mediated by the enzyme carnitine palmitoyltransferase-1 (CPT1) [87]. Atenolol and Enalapril are both antihypertensive medications. Atenolol is a hydrophilic beta-receptor blocking drug, while Enalapril is a long-acting, nonsulfhydryl Angiotensin-Converting Enzyme (ACE) inhibitor [88,89]. Dextromethorphan (DXM), which is commonly used as an antitussive medication, acts as a non-competitive, low-affinity antagonist of the NMDA receptor and exhibits potential neuroprotective properties [77]. Prazosin is a selective α1-adrenergic antagonist, promoting the idea that theoretical models and preclinical findings support the involvement of noradrenergic circuits in alcohol reinforcement and the propensity for relapses [84]. Oxytocin is a neuropeptide that is widely distributed within the brain and alters neuroadaptation to addictive substances, including alcohol [82].

Ondansetron, a highly selective 5HT3 receptor antagonist, has been shown to reduce cravings when it is combined with Naltrexone [24]. In one study, where it was used as monotherapy, Ondansetron significantly reduced the overall cravings among early-onset alcoholics. In contrast, it increased the cravings in late-onset alcoholics [18].

The highlight of those was Bromocriptine, which in the two studies that were evaluated, showed a positive result for cravings reduction [11,79]. Acetyl-L-carnitine and Intranasal Oxytocin also showed cravings reduction in the only studies that were found [80,82]. Prazosin was focused on in two studies; one of them showed a better result for cravings [75], while in the other study, its use did not differ from that of the placebo [84]. The other drugs did not show a positive result regarding abstinence [74,76,77,78,81,84].

As for abstinence, Bromocriptine also had a favorable response in two articles that were studied [11,79]. Enalapril showed a more favorable result for abstinence compared to that of the placebo at a dose of 20 mg/day, but did not differ at a dose of 10 mg [81]. Acetyl-L-carnitine showed a good response to abstinence [80]. In the studies of the other drugs, abstinence was not evaluated or not shown to benefit the patients more than the placebo did.

In contrast to all the studies that used pharmacological interventions, some used Kudzu, a climbing vine native to Asia. The roots, flowers, and leaves are used as medicine in China to treat variety of disorders, including neck pain, eye pain, fever, and measles. More recently, Chinese people have used Kudzu in the treatment of alcohol addiction. In only one study, there was no difference between the use of this and the placebo in terms of maintaining abstinence and cravings reduction [83].

## 4. Discussion

The treatment of AUD remains one of the greatest challenges in psychiatry due to the scarcity of specific and recent treatments as well as the difficulty in carrying out long-term studies with large samples due to the complexity of the population that serves as a sample. The main objective of this review was to list the possible treatments for alcohol cravings, but also to assess whether there was a satisfactory long-term response in maintaining abstinence. The drugs with the highest number of studies were Naltrexone (thirteen trials; eight monotherapy and five combined with other interventions), Baclofen (six), and Acamprosate (six). Among the eight trials in which Naltrexone was tested alone, there was a favorable response to reducing cravings in seven trials. However, when Naltrexone was associated with another intervention, it did not show a reduction in craving in four out of five analyzed studies. Acamprosate and Baclofen, in monotherapies, showed positive results in reducing cravings in two out of five and three out of six trials, respectively.

Eleven different substances associated with a few studies were beneficial in all the trials: Varenicline (two), Bromocriptine (two), Gabapentin (two), Acetyl-L-carnitine (one), Fluoxetine (one), Gamma-hydroxybutyric Acid (one), Quetiapine (one), Samidorphan (one), Ondansetron (one), and Intranasal Oxytocin. Although these drugs are widely used in psychiatry, there were only a few studies. This demonstrates the need for more targeted Randomized Controlled Trials (RCT) to assess the cravings reduction.

Among the specific drugs used for the treatment of alcoholism, Naltrexone was still the best option for reducing alcohol cravings, followed by Acamprosate. In one study where Naltrexone, Acamprosate, and a placebo were evaluated, only Naltrexone showed a reduction of cravings, while Acamprosate had no effect. When it was combined with other therapies, such as PUFAS (Polyunsaturated Fatty Acids) and Cognitive Behavioral Therapy, Naltrexone did not differ from the placebo [25,27,28]. However, when Naltrexone was associated with psychotherapeutic approaches, such as CBT and/or ST, there was a more positive response in terms of abstinence compared to that of the placebo associated with these interventions. This demonstrates how much more Naltrexone can contribute to multidisciplinary treatments compared to that of isolated psychotherapy.

When combined with other pharmacological interventions, Naltrexone plus Acamprosate were superior to the placebo in maintaining abstinence, while, when they were associated with Ondansetron, it promoted cravings reduction [24,33].

To promote abstinence, Naltrexone, Acamprosate, and Nalmefene, acting in a monotherapy, were effective in 30% (three) of the studies, showing that there are beneficial results not only to attenuate the symptoms of craving, but also to maintain long-term abstinence.

Out of the anticonvulsants, Gabapentin is one of the drugs widely used in the treatment of dependences; it has shown good results both in reducing cravings and maintaining abstinence among alcohol-dependent patients. Topiramate, GHB, Pregabalin, and Levetiracetam were present in a few studies, but demonstrate some results both for abstinence and for reducing cravings in relation to the placebo. This class of drugs thus represents a possible second line of treatment for AUD.

Baclofen, which is already a drug used by those with alcoholism, especially those who suffer from a certain chronic liver disease resulting from alcohol use, showed promising results in the five articles that were found in this review, with cravings reduction in two and favorable abstinence results in three. This drug may be used as an adjuvant due to these favorable results, with only a few side effects; this drug may be used as an as a therapeutic option due to its favorable results with, only a few side effects, as well as Gabapentin, especially for patients with already compromised liver function.

Varenicline, which is a drug widely used in the treatment of smoking cessation, had a significant impact. Two studies had a 100% response rate to cravings, which was compared to that of the placebo, for those who use tobacco and alcohol at the same time; for those with dependences, there was one positive result for abstinence for both substances.

Furthermore, psychotropic drugs that are frequently used in the treatment of AUD, despite there only being a few studies, only Fluoxetine and Quetiapine were beneficial in reducing cravings; the latter was also effective for abstinence. Such medications could perhaps be used as adjunctive treatments for those with addictions, especially in patients with psychiatric comorbidities.

Of the drugs that are not frequently associated with the treatment of mental disorders, including AUD, Bromocriptine, a well-known dopamine agonist, which is used more frequently to treat Parkinson’s disease, showed a favorable result in terms of both abstinence and cravings, as well as Acetyl-L-carnitine. Additionally, Prazosin was included in one study, showing a positive result in reducing cravings. Such drugs could perhaps be used as a third-line treatment when all other alternatives have been used.

Of the 60 articles, 47 assessed both the outcomes of cravings reduction and abstinence. Six articles only analyzed cravings. Of the 47 studies, 23 articles showed that abstinence was favored by the control group, and 29 articles showed that there was a larger reduction of cravings in the control group compared to that of the placebo group.

Fourteen studies were successful in both reducing cravings and maintaining abstinence, including those on: Naltrexone (two out of seven studies), Acamprosate (one out of four), Gabapentin (two out of three), Topiramate (two out of three), Baclofen (one out of six), Bromocriptine (two out of two), Acetyl-L-carnitine (one out of one), Quetiapine (one out of one), Ondansetron (one out of one), Varenicline (in two studies where both outcomes were evaluated, it was positive in both in only one, and in the other article, it was positive for craving reduction and showed no difference compared to the placebo in maintaining abstinence).

Regarding the specific drugs used for an alcohol abuse treatment, Naltrexone proved to be quite effective in reducing cravings (85.7%); however, the authors obtained worse abstinence results (33.33%) for both criteria (28.5%). Acamprosate, which only showed the benefits in both abstinence and cravings reduction in one article, did not have a positive result for both criteria in the other two studies found. Nalmefene, on the other hand, did not show a significant difference compared to the placebo in one article.

About anticonvulsants, from 14 articles found where abstinence and cravings reduction were evaluated, five articles showed benefits in both criteria, while eight included the cessation of abstinence. Interestingly, Baclofen resulted in the patients comprising a higher proportion of the abstinence maintenance group (three out of six articles), but a lower proportion of the cravings reduction group (two out of six). Both Gabapentin and Topiramate were focused on in two studies each with consistent results in promoting abstinence. Additionally, there were three studies on each drug that demonstrate concordant results in reducing cravings. However, for Topiramate, out of the three studies, only two evaluated abstinence, and these showed favorable results in compared to those of the placebo.

Out of the fourteen studies involving anticonvulsant drugs, eight obtained positive results for abstinence maintenance. The results were better than those observed for specific drugs used for the treatment of alcoholism, which resulted in a higher proportion of patients with cravings reduction.

Furthermore, psychotropic drugs are frequently employed in the treatment of AUD, despite there only being a few studies; only Fluoxetine, Samidorphan, and Quetiapine were beneficial in reducing cravings; the latter was also effective for abstinence. These medications may hold potential as adjunctive treatments, especially for patients with psychiatric comorbidities.

Among the drugs not typically associated with mental disorders or AUD treatment, Bromocriptine, a well-known dopamine agonist, used primarily to treat Parkinson’s disease, exhibited favorable results in both abstinence and cravings reduction, as well as Acetyl-L-carnitine. Additionally, one study demonstrated the positive effect of Prazosin in reducing cravings. These drugs could perhaps be used as a third-line treatment option when all other alternatives have been exhausted.

Despite providing valuable insights, the analyzed studies have certain limitations. Firstly, there is a paucity of research on psychotropic drugs for AUD. Secondly, not all studies that evaluated cravings also assessed abstinence, making it difficult to establish a correlation between these two outcomes. Thirdly, there is a substantial disparity in the study durations with some being long, and other being short. Furthermore, approximately one-quarter of the studies did not report the number of dropouts, and a publication bias may have influenced the reported outcomes. Moreover, the heterogeneity of the cravings measures precluded a meta-analysis of standardized instruments. Lastly, this review relied on articles from English language databases, and no studies from African and Middle Eastern countries were found, possibly due to there being limited research in these regions and less alcohol consumption in Muslim countries.

## 5. Conclusions

In conclusion, this review underscores the ongoing challenges in treating AUD and highlights the potential of various medications in reducing cravings and maintaining long-term abstinence. Naltrexone and Acamprosate emerged as the most effective options, with other substances showing promise, but requiring future investigation through targeted RCTs. Additionally, anticonvulsants and certain psychotropic drugs may offer viable treatment alternatives. However, it is crucial to address the limitations of the mentioned studies and conduct comprehensive research to advance our understanding of cravings and optimize their assessment and treatment approaches.

Despite the fact that specific drugs used for the treatment of alcoholism are the gold standard for reducing cravings (especially Naltrexone), not all studies have shown a good response for maintaining abstinence. About 1/3 of the studies demonstrated a reduction of cravings and the maintenance of abstinence compared to those of the placebo.

As for anticonvulsants, more than half of the studies had a favorable result in the cessation of cravings and in maintaining abstinence. A total of 57% of studies using anticonvulsants obtained a favorable response to abstinence. A possible therapeutic alternative would be not only to switch specific drugs for the treatment of alcoholism, such as Naltrexone, to anticonvulsants, but also to associate them for refractory cases or situations where the expected response was not achieved, particularly when the goal is to reach total abstinence.

There are promising results with anti-smoking drugs for those who suffer from a comorbidity associated with alcohol and tobacco dependence. Psychotropics, such as antidepressants and antipsychotics, and drugs that are not regularly used for the treatment of AUD could be used as a third line for a possible treatment of some cases. Such a review may direct the development of a possible algorithm for the treatment of alcoholism in the future and open perspectives for future studies on the aforementioned medications, further increasing the level of evidence for each treatment.

## Figures and Tables

**Table 1 brainsci-13-01206-t001:** Specific drugs used for the treatment of alcoholism.

Author, Year, Country	Sample: *n*; Age (SD or Range); Male:Female; Dependence	Setting	Measures	Measures Craving	Interventions	Follow-Up	Outcomes Abstinence	Outcomes Craving	Dropout	Citation
Ait-Daoud, N., et al. (2001) [24]; USA	20; 38.0 ± 1.78; males 75%; DSM-4 criteria	Inpatient	TLFB; Michigan Alcoholism Screening Test (MAST) and CIWA-Ar	OCDS	Ondansetron + Naltrexone	8 weeks	Was not Analyzed	Reduces craving than placebo	0	5
Anton, R.F., et al. (1999) [25]; USA	131; Naltrexone: 41 ± 10, placebo: 44 ± 10; Naltrexone: males 69%, placebo: males 73%; DSM-3 criteria	Outpatient	Addiction Severity Index (ASI); ADS; Form 90; CDT and GGT	Four analog scales measuring craving and OCDS	CBT + Naltrexone	12 weeks	Favorable in control group	Do not differ from placebo	17.56%	7
Anton, R.F., et al. (2004) [26]; USA	270; Placebo: 45 ± 11, Nalmefene 5 mg: 45 ± 11, 20 mg: 46 ± 11, 40 mg: 44 ± 9; males: 71.8%; DSM-4 criteria	Outpatient	CIWA-Ar; SCID; TLFB; ADS; DrInC; CDT and GGT	OCDS	Nalmefene	12 weeks	Non-significant reduction in craving than placebo	Non-significant reduction in craving than placebo	25.90%	8
Balldin, J., et al. (2003) [27]; Sweden	118; CBT + Naltrexone: 50 ± 7, ST + Naltrexone: 48 ± 8, CBT + Placebo: 50 ± 8, ST + Placebo: 51 ± 8; males: CBT + Naltrexone: 84%, ST + Naltrexone: 87%, CBT + Placebo: 77%, ST + Placebo: 91%; DSM-4 criteria	Outpatient	ASI; CDT; SCL-90 and TLFB	OCDS	CBT + Naltrexone; ST + Naltrexone	6 months	Favorable in control group	Non-significant reduction in craving than placebo	23.00%	9
Chick, J. et al. (2000) [5]; USA	169; placebo: 43.9 ± 9.7, naltrexone: 43.1 ± 8.3; males: 74.8%; DSM-3 criteria	Outpatient	TLFB; ASI	OCDS	Naltrexone	12 weeks	Do not differ from placebo	Reduces craving than placebo	58.50%	12
Fogaça, M.N., et al. (2011) [28]; Brazil	80; aged between 30 and 50; males 100%; DSM-4 criteria	Outpatient	The Short Alcohol Dependence Data (SADD)	Obsessive Compulsive Drinking Scale (OCDS)	PUFAS; Naltrexone; Naltrexone + PUFAS	90 days	Was not Analyzed	Do not differ from placebo	46.25%	18
Garbutt, J.C., et al. (2016) [6]; USA	80; 47.0 ± 8.6; males 71.25%; DSM-4 criteria	Outpatient	MINI	PACS	Naltrexone	12 weeks	Favorable in control group	Reduces craving than placebo	20.00%	23
Gastpar, M., et al. (2002) [29], Germany	171; 42.7 ± 9.7; males 72.5%; DSM-3 criteria	Outpatient and inpatient	TLFB; GGT; AST and ALT	OCDS and the ASI-craving	Naltrexone	12 weeks	Non-significant reduction in craving than placebo	Non-significant reduction in craving than placebo	35.67%	24
Guardia, J., et al. (2002) [7]; Spain	192; Naltrexone: 41 ± 8, Placebo: 42 ± 9, males: naltrexone 72%, placebo 77%; DSM-4 criteria	Outpatient and Inpatient	Biological markers of heavy drinking (MCV, GGT, and CDT), and markers of possible toxicity (AST, ALT, and thrombocytes)	11-point Likert scale from null desire (0) to irresistible desire (10)	Naltrexone	12 weeks	Do not differ from placebo	Reduces craving than placebo	40.63%	27
Hammarberg, A., et al. (2009) [30]; USA	56; acamprosate 50.2 ± 7.6, placebo 49.8 ± 7.3; males 53.57%; DSM-4 criteria	Outpatient	Time-Line Follow Back (TLFB), Beta-endorphin, ACTG, Cortisol	Desire for alcohol Questionnaire (Short DAQ); OCDS	Acamprosate	21 days	Was not Analyzed	Reduces craving than placebo	N/A	29
Higuchi, S., et al. (2015) [31]; Japan	327; acamprosate: 51.7 ± 12.4, placebo: 53.1 ± 12.2; males: acamprosate 86.5%, placebo 88.4%; ICD-10 criteria	Inpatient	GGT; ALT and AST	Craving questionnaire	Acamprosate	24 weeks	Favorable in control group	Reduces craving than placebo	7.10%	30
Huang, M.C., et al. (2005) [32]; Taiwan	40; naltrexone: 38.1 ± 5.8, placebo: 42.9 ± 9.3; males 92.5%; DSM-3 criteria	Outpatient	AST, ALT and GGT	Self-rating craving scores	Naltrexone	14 weeks	Do not differ from placebo	Reduces craving than placebo	40.00%	31
Kiefer, F., et al. (2005) [33]; Germany	160; 46.2 ± 9.3; males 73.7% DSM-4 criteria	Inpatient	Breath Alcohol concentration (BrAC) + Urinary drug screens; GGT; MCV; CDT and Leptin plasma concentration during treatment	VAS and self-rated craving	Naltrexone; Acamprosate; Naltrexone + Acamprosate	12 weeks	Abstinence: Favorable in control group (Acamprosate plus Naltrexone)	Do not differ from placebo	31.25%	44
Monterosso, J.R., et al. (2001) [34]; USA	183; 46.2 ± 11.5; males 72.8%; DSM-3 criteria	Outpatient	ASI; TLFB and PACS	PACS	Naltrexone	12 weeks	Was not Analyzed	Reduces craving than placebo	17.90%	64
Morley, K.C., et al. (2006) [35]; Australia	169; 45 ± 9; males 70%; DSM-4 criteria	Inpatient	Alcohol Dependence Scale (ADS) and Depression Anxiety and Stress Scale (DASS)	PACS	Naltrexone; Acamprosate	12 weeks	Do not differ from placebo	Naltrexone: Reduces craving than placebo Acamprosate: do not differ from placebo	30.77%	66
Namkoong, K., et al. (2003) [36]; South Korea	142; 44.3 ± 8.3; males 95.8%; DSM-4 criteria	Outpatient and Inpatient	TLFB	OCDS and VAS	Acamprosate	8 weeks	Do not differ from placebo	Do not differ from placebo	28.87%	67
Umhau, J.C., et al. (2011) [37]; USA	35; 44.4 (1.6–1.9); males 88%; DSM-4 criteria	Inpatient	Fagerstöm; ASI; ADS; TLFB; CIWA-AR and STAI	PACS and AUQ	Acamprosate	2 weeks	Was not Analyzed	Do not differ from placebo	28.50%	86
Volpicelli, J.R., et al. (1992) [38]; USA	70; placebo: 43.3 ± 9, naltrexone: 43.5 ± 9.3; males 100%; DSM-3 criteria	Outpatient	MAST; 90-Item Symptom Checklist (SCL-90); Brief Psychiatric Rating Scale (BPRS); AST and GGT and number of drinking days.	Self-rating craving scores	Naltrexone	12 weeks	Favorable in control group	Reduces craving than placebo	35.70%	87

SD—Standard Deviation; DSM—Diagnostic and Statistical Manual of Mental Disorders; CBT—Cognitive Behavioral Therapy; ST—Self-Test; ICD—International Classification of Diseases; TLFB—Timeline Follow-Back; CIWA-Ar—Clinical Institute Withdrawal Assessment for Alcohol, Revised; ADS—Alcohol Dependence Scale; CDT—Carbohydrate-Deficient Transferrin; SCID—Structured Clinical Interview for DSM Disorders; GGT—Gamma-Glutamyl Transferase; ASI—Addiction Severity Index; SCL—Symptom Checklist; MINI—Mini International Neuropsychiatric Interview; ALT—Alanine Aminotransferase; MCV—Mean Corpuscular Volume; CTG—AIDS Clinical Trials Group; PACS—Penn Alcohol Craving Scale; STAI—State-Trait Anxiety Inventory; MAST—Michigan Alcohol Screening Test; OCDS—Obsessive-Compulsive Drinking Scale; DAQ—Desires for Alcohol Questionnaire; VAS—Visual Analog Scale; AUQ—Alcohol Use Questionnaire; PUFAS—Polyunsaturated Fatty Acids.

**Table 2 brainsci-13-01206-t002:** Anticonvulsants.

Author, Year, Country	Sample: *n*; Age (SD or Range); Male:Female; Dependence	Setting	Measures	Measures Craving	Interventions	Follow-Up	Outcomes Abstinence	Outcomes Craving	Dropout	Citation
Addolorato, G., et al. (2002) [43]; Italy	39; 47.3 ± 10.5; DSM-4 criteria	Outpatient	Self-evaluation; Family member interview; determination of alcohol concentration in blood and saliva by QED; Cumulative Abstinence duration (CAD)	OCDS	Baclofen	30 days	Favorable in control group	Reduces craving than placebo	30.76%	3
Falk, D.E., et al. (2019) [13]; USA	346; placebo: 49.4 ± 11.4, GE-XR: 50.7 ± 10.3; males 66%; DSM-5 criteria	Outpatient	TLFB; 90 interview; WHO drinking risk categories, ImBIBe, Pittsburg Sleep Quality Index (PSQI), Beck Anxiety Inventory, Beck Depression Inventory Scale–II, POMS	Alcohol Craving Scale–Short Form	Gabapentin enacarbil extended-release	6 months	Do not differ from placebo	Do not differ from placebo	15.00%	17
Furieri, FA., et al. (2007) [14]; Brazil	60; placebo: 43.87, gabapentin: 44.67; males 100%; DSM-4 criteria	Outpatient	DSM-IV; age of drinking onset; self-reported drinking over the past 90 days by TLFB and UKU	OCDS and CIWA-Ar	Gabapentin	4 weeks	Favorable in control group	Reduces craving than placebo	20.00%	20
Gallimberti, L., et al. (1992) [15]; Italy	82; placebo: 36.8 ± 15.6, GHB: 38.1 ± 13.4; males 66.2%; DSM-3 criteria	Outpatient	S-GGT; E-NCV; self-reported alcohol consumption; alcoholuria test; VAST; Spielberger’s State and Trait Anxiety Scale; Hamilton Depression Scale	Stunkard and Messick’s Questionnaire	Gamma-Hydroxybutyric Acid	3 months	Do not differ from placebo	Reduces craving than placebo	13.41%	21
Garbutt, J.C., et al. (2010) [42]; USA	80; placebo: 50.3 ± 7.2, baclofen: 47.5 ± 7.6; males 55%; DSM-4 criteria	Outpatient	SCID, MINI, TLFB, ADS, Zung Rating Depression Scale, STAI, CIWA-Ar	Pennsylvania Alcohol Craving Scale (PACS)	Baclofen	12 weeks	Do not differ from placebo	Do not differ from placebo	24.00%	22
Johnson, B.A., et al. (2003) [16]; USA	150; topiramate: 41.51± 8.75, placebo: 42.05 ± 8.83; males 71.33%; DSM-4 criteria	Outpatient	Haematological and biochemical laboratory studies, urine drug test; BAC and TLFB	OCDS	Topiramate	12 weeks	Favorable in control group	Reduces craving than placebo	31.33%	39
Koethe, D., et al. (2007) [44]; Germany	50; 46.4 ± 7.4; males 80%; DSM-4 criteria	Inpatient	SAB score; European Addiction Severity Index; The Münchner Alkoholismus Test (MALT); STAI; BDI; Brief Symptom Inventory of Derogatis (BSI); self-assessment questionnaire SCL-90; AES scale; Pittsburgh Sleep Quality Index (PSQI).	VAS	Oxcarbazepine	6 days	Was not Analyzed	Do not differ from placebo	1%	46
Krupitsky, E.M., et al. (2015) [45], Russia	32; baclofen: 46 ± 2.43, placebo: 44 ± 2.12; males 81.25%; ICD-10 criteria	N/A	TLFB; Global Clinical Impression Scale (GCI), Hamilton, Spielberger; GGT; Montgomery-Ashberg (MADRS)	OCDS; PACS and VAS	Baclofen	3 months	Do not differ from placebo	Reduces craving than placebo	12.50%	47
Krupitsky, E.M., et al. (2020) [46], Russia	100; pregabalin: 43 ± 1.27, placebo: 45.92 ± 1.4; males 83%; ICD-10 criteria	N/A	TLFB; Global Clinical Impression Scale (GCI), Hamilton, Spielberger, MADRS	Visual Analog Scale (VAS)	Pregabalin	24 weeks	Favorable in control group	Do not differ from placebo	50.00%	48
Likhitsathian, S., et al., (2013) [2], Thailand	106; 41.55; males 100%; DSM-4 criteria	Inpatient	AUDIT score 8 or more; GGT; Mini International Neuropsychiatric Interview (MINI) and CIWA-Ar	Visual Analog Technique	Topiramate	12 weeks	Was not Analyzed	Do not differ from placebo	50.00%	52
Mason, B.J., et al. (2014) [47]; USA	150; placebo: 46.8 ± 11.3, gabapentin 900 mg: 41.9± 10.1 and 1800 g: 45.2 ± 11.3; males 56.7%; DSM-4 criteria	Outpatient	SCID, TLFB, Beck Depression Inventory II; CIWA-AR	Alcohol Craving Scale–Short Form	Gabapentin	24 weeks	Favorable in control group	Reduces craving than placebo	43.30%	58
Morley, K.C., et al. (2018) [48]; Autralia	72; placebo: 50 ± 11.2, baclofen 30–75 mg: G-: 46.7 ± 9.7, CC: 49.7 ± 7.8, G-: 47.9 ± 9.8, No DNA: 47.7 ± 10.9; males 70.2%; ICD-10 criteria	Outpatient and Inpatient	TLFB; GGT, AST and ALT	PACS	Baclofen	12 weeks	Favorable in control group	Non-significant reduction in craving than placebo	31.00%	65
Ponizovsky, A.M., et al. (2014) [49]; Israel	64; placebo: 44.7 ± 8.7, baclofen: 42.6 ± 9.6; males 75%; ICD-10 criteria	Outpatient	Proportion of Heavy Drinking Days (%HDD); the proportion of total abstinent days (%ABS); TLFB; BDI; General Health Questionnaire (GHQ-12); General Self-Efficacy Scale (GSES); Multidimensional Scale of Perceived Social Support (MSPS); Quality of Life Enjoyment and Satisfaction Questionnaire (Q-LES-Q)	OCDS	Baclofen	12 weeks	Do not differ from placebo	Reduces craving than placebo	38.00%	73
Richter, C., et al. (2012) [3]; Germany	242; placebo: 48.1 ± 9.1, Levetiracetam: 47.3 ± 9.9; males 71.5%; DSM-4 criteria	Outpatient	TLFB; drug urine for benzodiazepines or other sedative hypnotics test and breath alcohol test	OCDS	Levetiracetam	16 weeks	Do not differ from placebo	Do not differ from placebo	16.90%	75
Rombouts, A.S., et al. (2019) [50]; Australia	104; age 18–75; NA; ICD-10 criteria	Inpatient	DASS; Drinks per drinking day; Drinks per drinking day; TLFB and CIWA-AR	PACS	Baclofen	12 weeks	Favorable in control group	Do not reduce craving than Placebo	0	76
Rubio, G., et al. (2009) [51]; Spain	76; topiramate: 42.5 ± 9.31, placebo: 42.07 ± 8.73; males 100%; DSM-4 criteria	Outpatient	CIWA-AR; Barratt Impulsiveness Scale; The Hamilton Anxiety Scale; Hamilton Depression Scale and carbohydrate-deficient transferrin (CDT)	3 analogous scales to measure the frequency, duration and intensity of craving	Topiramate	12 weeks	Abstinence: Favorable in control group	Reduces craving than placebo	17.10%	77

GE-XR: Gabapentin enacarbil extended release; GHB—Gamma-Hydroxybutyrate; S-GGT—Serum Gamma-Glutamyl Transferase; MINI—Mini In-ternational Neuropsychiatric Interview; STAI—State-Trait Anxiety Inventory; BAC—Blood Alcohol Concentration; SAB Score—German; score-directed treatment of alcohol withdrawal syndrome; AES—Alcohol Expectancy Scale; MADRS—Montgomery-Åsberg Depression Rating Scale; AUDIT—Alcohol Use Disorders Identification Test; BDI—Beck Depression Inventory; GHQ—General Health Questionnaire; DASS—Depression Anxiety Stress Scales.

**Table 3 brainsci-13-01206-t003:** Varenicline and Mecamylamine.

Author, Year, Country	Sample: *n*; Age (SD or Range); Male:Female; Dependence	Setting	Measures	Measures Craving	Interventions	Follow-Up	Outcomes Abstinence	Outcomes Craving	Dropout	Citation
de Bejczy, A., et al. (2015) [12]; Sweden	160; placebo: 55.6, Varenicline: 54.6; males 62%; DSM-4 criteria	Outpatient	Specific alcohol marker phosphatidylethanol (PEth) in the blood; Alcohol Use Disorders Identification Test (AUDIT)	OCDS	Varenicline	12 weeks	Do not differ from placebo	Reduces craving than placebo	11.70%	15
Litten, R.Z., et al. (2013) [61]; USA	200; Placebo: 45 ± 12.3, Varenicline: 46 ± 11; males 71%; DMS-4 criteria	Outpatient and inpatient	TLFB	PACS	Varenicline	13 weeks	Favorable in control group	Reduces craving than placebo	14.80%	53
Petrakis, I.L., et al. (2018) [62]; USA	128; 48.5 ± 9.4; males 85.9%; DSM-4 criteria	Outpatient	Alcohol Dependence Severity (ADS); the Drinker Inventory of Consequences (DrInC); TLFB; Fagerstrom Test for Nicotine Dependence (FTND); GGT, CDT; Questionnaire of Smoking Urges (QSU); Systematic Assessment for Treatment Emergent Events (SAFTEE)	OCDS	Mecamylamine	12 weeks	Do not differ from placebo	Non-significant reduction in craving than placebo	36.90%	72

**Table 4 brainsci-13-01206-t004:** Other psychotropic drugs.

Author, Year, Country	Sample: *n*; Age (SD or Range); Male:Female; Dependence	Setting	Measures	Measures Craving	Interventions	Follow-Up	Outcomes Abstinence	Outcomes Craving	Dropout	Citation
Evans, S.M., et al. (2007) [70]; USA	44; placebo: 42.5 ± 11.6, memantine: 42.6 ± 8.6; males 79.4%; DSM-4 criteria	Outpatient	TLFB; GGT; self-reported alcohol use; BDI	OCDS	Memantine	16 weeks	Favorable in placebo group	Non-significant reduction in craving than placebo	22.72%	16
Guardia, J., et al. (2004) [71]; Spain	60; olanzapine: 42.52 ± 10.16, placebo: 44.32 ± 13.74; males 76.66%; DSM-4 criteria	Outpatient	ASI; ADS; Impulsivity (KPS); Psychiatric symptoms (SCL-90R); Stages of change (URICA); Arger (STAXI); Paranoidism; TLFB; BDI; Anxiety symptoms (STAI-E); Global Clinical Impression (GCI); Extrapyramidal symptoms (ECU)	VAS; AUQ	Olanzapine	12 weeks	Do not differ from placebo	Do not differ from placebo	31.66%	28
Johnson, B.A., et al. (1996) [17]; USA	423; 41.2; males 77.3%; DSM-3 criteria	Inpatient	Addiction Severity Index (ASI) and Short Michigan Alcohol Screening Test (SMAST)	VAS	Ritanserin	12 weeks	Was not Analyzed	Do not differ from placebo	38.30%	38
Kabel, D.I., et al. (1996) [72]; USA	33; 46.8 ± 9.1; males 100%; DSM-3 criteria	Inpatient	MAST; Questionnaire on Drinking Urges; Questionnaire on Smoking Urges	10 point-scale	Fluoxetine	12 weeks	Do not differ from placebo	Reduces craving than placebo	15.15%	40
Kampman, K.M., et al. (2007) [73]; USA	61; 47 ± 8.8; male: 77% × female 23%; DSM-4 criteria	Outpatient	SCID, CIWA-Ar, HAM-D, 3 or more childhood criteria for Antisocial Personality Disorder, TLFB, ASI	PACS	Quetiapine	12 weeks	Favorable in control group	Reduces craving than placebo	23.00%	41
Malec, E., et al. (1996) [68]; Canada	57; 41.63; males 82.4%; DSM-3 criteria	Outpatient	Michigan Alcoholism Screening Test, Alcohol Use Inventory, Drinking Behaviour Intewiew (DBI), HAM-A, MADRS, SCL-90-R	VAS	Buspirone	12 weeks	Do not differ from placebo	Do not differ from placebo	36.84%	55
Marra, D., et al. (2002) [67]; France	72; 43.5; males 69.44%; DSM-4 criteria	Inpatient	OCDS and a self-report evaluation of craving that includes an obsessive subscale and TLFB	VAS	Amisulpride	6 months	Favorable in placebo group	Non-significant reduction in craving than placebo	55.55%	57
O’Malley, S.S., et al. (2018) [66]; USA	410; 42.9 ± 11.32; males 71%; DSM-4 criteria	Outpatient	Percentage of Subjects with No Heavy Drinking Days (PSNHDD); TLFB; Patient Global Assessment of Response to Therapy (PGART)	Patient’s ratings of alcohol craving and VAS	Samidorphan	12 weeks	Do not differ from placebo	Reduces craving than placebo	38.29%	69
Wiesbeck, et al. (1999) [69]; Germany	493; 42.6; males 80.5%; DSM-3 criteria	Outpatient	Withdrawal Syndrome Scale for Alcohol	VAS	Ritanserin	6 months	Do not differ from placebo	Do not differ from placebo	45.23%	90
Wong, W.M., et al. (2008) [65]; Germany	23; 36.45 ± 7.7; males 100%; DSM-4 and ICD-10 criteria	Inpatient	ACTH Blood Levels; Semi-structured interview for assessment of genetics in alcoholism (SSAGA)	VAS	Citalopram	1 day	Was not Analyzed	Do not differ from placebo	0	91

KPS—Karnofsky Performance Status; URICA—University of Rhode Island Change Assessment; STAXI—State-Trait Anger Expression Inventory; STAI-E—State-Trait Anxiety Inventory-Estado; HAM-D—Hamilton Depression Rating Scale; MADRS—Montgomery-Åsberg Depression Rating Scale; ACTH—Adrenocorticotropic Hormone.

**Table 5 brainsci-13-01206-t005:** Other drugs.

Author, Year, Country	Sample: *n*; Age (SD or Range); Male:Female; Dependence	Setting	Measures	Measures Craving	Interventions	Follow-Up	Outcomes Abstinence	Outcomes Craving	Dropout	Citation
Alho, H., et al. (2002) [74]; Finland	105; N/A.	Inpatient	Heart rate, blood pressure, pulse oximetric saturation, frontal muscle electromyographic activity, and plethysmographic pulse amplitude; liver enzymes and severity of dependency with Severity of Alcohol Dependence Data	OCDS	Nitrous oxide/oxygen (from 30 to 70% in oxygen); air/oxygen (30%/70%); medical (normal) air	3 and 6 months	Was not Analyzed	Do not reduce craving than Placebo	N/A	6
Borg, V. (1983) [11]; Norway	50; group: 40.2 ± 11, placebo: 42.6 ± 7; males 82%; N/A	Outpatient	Drug-Taking Evaluation Scale (DTES)	Self-report	Bromocriptine	6 months	Favorable in control group	Reduces craving than placebo	40.00%	10
Fox, H.C., et al. (2012) [75]; USA	17; placebo: 36.11 ± 8.24, prazosin: 35.75 ± 7.67; males 70.6%; SCID-4 criteria	Outpatient	Differential Emotion Scale (DES); Cardiovascular Measure; Blood Sample (ACTH and cortisol samples)	VAS	Prazosin	4 weeks	Was not Analyzed	Reduces craving than placebo	0	19
Gottlieb, L.D., et al. (1994) [76]; USA	100; atenolol: 39 ± 13, placebo: 39 ± 14; N/A; Clinical evaluation criteria	Inpatient	Alcohol Dependence Questionnaire (SADQ)	5-point Likert scale	Atenolol	12 months	Do not differ from placebo	Do not differ from placebo	85.00%	26
Huang, M.C., et al. (2014) [77]; Taiwan	40; DXM: 42.3 ± 6.9, placebo: 38.2 ± 7.1; males 87.5%; DSM-4 criteria	Inpatient	CIWA-Ar	OCDS	Dextromethorphan	7 days	Was not Analyzed	Do not differ from placebo	0	31
Johnson, B.A., et al. (2002) [18]; USA	253; 41.45; males 70%; DSM-3 criteria	Outpatient	TLFB; Michigan Alcoholism Screening Test	VAS	Ondansetron	12 weeks	Favorable in control group	Reduces craving than placebo	N/A	39
Kenna, G.A., et al. (2016) [78]; USA	41; doxazosin: 42.1 ± 10.2, placebo: 42.1 ± 7.5; males 71%; N/A	Outpatient	TLFB; drinks per week and Heavy drinking days; Perceived Stress Scale (PSS); Hamilton Anxiety Scale; Anxiety-Tension Subscale of the Profile of Mood States (POMS-TA)	OCDS	Doxazosin	10 weeks	Do not differ from placebo	Non-significant reduction in craving than placebo	26.82%	42
Lawford, B.R., et al. (1995) [79]; USA	83; 43.7 ± 1.3; males 94%; DSM-3 criteria	Inpatient	Spielberger State Anxiety Inventory; BDI; Taql A DRD2 alleles	The Borg Craving Scale	Bromocriptine	6 weeks	Favorable in control group	Reduces craving than placebo	37.00%	49
Martinotti, G., et al. (2010) [80], Italy	64; group 1: 44.9 ± 13.8, group 2: 38.6 ± 8.47, placebo: 42.7 ± 12.3; males 56.25%; DSM-4 criteria	Outpatient	SCID 1 and 2; ASI; SHAPS; CIWA-Ar	Visual Analogue Scale (VAS)	Acetyl-L-carnitine	142 days	Favorable in control group	Reduces craving than placebo	57.80%	60
Naranjo, C.A., et al., (1991) [81]; Canada	141; enalapril: 40.7 ± 10, placebo: 43.5 ± 9.3; males 85.7%; DSM-3 criteria	Outpatient	TLFB; Social stability index; MAST; ADS; Spielberg’s anxiety scales; BDI; Montgomery-Asberg depression scale; AST; ALT; GGT; Plasma renin activity	Self-report	Enalapril (10 mg and 20 mg)	4 weeks	Favorable in control group (enalapril 20 mg superior than placebo, but, Enalapril 10 mg do not differ than placebo).	Do not differ from placebo	30.50%	68
Pedersen, C.A., et al. (2013) [82]; USA	11; 41.27 ± 14.67; males 81.81%; MINI criteria	Inpatients	Alcohol Withdrawal Symptom Checklist (AWSC); Clinical Institute Withdrawal Assessment for Alcohol (CIWA); POMS and TLFB	Penn Alcohol Craving Scale (PACS), Alcohol Craving Visual Analog Scale (ACVAS)	Intranasal oxytocin	3 days	Was not Analyzed	Reduces craving than placebo	0	71
Shebek, J., et al. (2000) [83]; USA	49; control: 48 ± 9, placebo: 48 ± 11; males 97.9%; DSM-IV criteria	Outpatient	N/A	VAS	Kudzu Root Extract	30 days	Do not differ from placebo	Do not differ from placebo	22.45%	78
Simpson, TL., et al. (2009) [84]; USA	24; 45.5 ± 8.0; males 79.1%; DSM-4 criteria	Outpatient	PSTD Checklist; 6-week Form-90; laboratory assessment of CBC, electrolytes, liver function panel, urine pregnancy and urine toxicology	PACS	Prazosin	6 weeks	Favorable in control group	Do not differ from placebo	17.00%	80

SADQ—Severity of Alcohol Dependence Questionnaire; SHAPS—Snaith-Hamilton Pleasure Scale; POMS—Profile of Mood States; CBC—Complete Blood Count.

## Data Availability

Not applicable.

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
