# Peer review of "Pharmacological Treatment of Alcohol Cravings"

_brainsci, 2023, doi:10.3390/brainsci13081206_

Round 1
Reviewer 1 Report
1. The authors do not mention findings from any of the existing meta-analyses of alcohol craving, of which there are several (some are listed below). Although some of these may be outside the focused scope of the article, which is pharmacological treatment of alcohol craving, it would be valuable to potential readers to at least briefly summarize the findings of similar meta-analyses of non-pharmacological treatments such as exercise and non-invasive brain stimulation.
Furthermore, one of the meta-analyses below (Blodgett et al., 2014) assessed the effect of topiramate on individuals with alcohol use disorders. Within the study, they explicitly examined the effect of topiramate on craving for 6 studies, all of which were RCTs. The authors’ current meta-analysis, which was conducted 9 years after the previous one, only found 2 topiramate studies (both of which were listed in the aforementioned meta-analysis). This makes me worried that the authors used too narrow a set of criteria not only for topiramate, but for all pharmacological treatments. Can the authors either argue why their methodology is more appropriate than that used in the Blodgett et al. (2014) study, or broaden their methodology so that it includes the studies analyzed in the Blodgett et al. (2014) study (as well as, hopefully, missing studies of other pharmacological agents)?
Blodgett JC, Del Re AC, Maisel NC, Finney JW. A meta‐analysis of topiramate's effects for individuals with alcohol use disorders. Alcoholism: Clinical and Experimental Research. 2014 Jun;38(6):1481-8.
Mostafavi SA, Khaleghi A, Mohammadi MR. Noninvasive brain stimulation in alcohol craving: A systematic review and meta-analysis. Progress in Neuro-Psychopharmacology and Biological Psychiatry. 2020 Jul 13;101:109938.
Hallgren M, Vancampfort D, Giesen ES, Lundin A, Stubbs B. Exercise as treatment for alcohol use disorders: systematic review and meta-analysis. British Journal of Sports Medicine. 2017 Jul 1;51(14):1058-64.
Hendershot CS, Wardell JD, Samokhvalov AV, Rehm J. Effects of naltrexone on alcohol self‐administration and craving: meta‐analysis of human laboratory studies. Addiction biology. 2017 Nov;22(6):1515-27.
2. The authors discuss some limitations at the end of the Discussion, one of which is “not all studies that evaluated craving, which is the main goal of this review, also evaluated abstinence (second goal). So, it is difficult to trace a correlation between the two outcomes adopted.” However, not all studies need to be included to run a correlation, just those that evaluated both abstinence and craving. Why can’t the authors assess a correlation on that subset of studies?
3. This brings up a similar point: Did the authors find a different line of treatment options for craving than for abstinence? In the introduction, the authors note that abstinence is the primary treatment outcome in most clinical trials, but that “an alternative to reducing alcohol consumption or even achieving abstinence would be to treat the craving.” The authors should therefore discuss if their findings suggest any difference in treatment approach for a craving compared to abstinence, or if any treatments seem to be more effective at treating craving compared to abstinence.
Minor points:
4. Near the beginning of the Results section, the authors state “Most studies use a population sample ranging from 16 and to 410 participants”. What about the other studies? When talking about a range you should include all studies in the analysis.
5. In the second paragraph of the Results, the authors mention the “Obsessive-Compulsive Disorder Scale”. Presumably, this should be the “Obsessive-Compulsive Drinking Scale”.
There were several minor grammatical errors in the manuscript that should be addressed, possibly by an editor well-versed in English.
Reviewer 2 Report
Pharmacological treatment of alcohol craving
Marin et al.,
This is a comprehensive review designed to identify pharmacological agents that have positive effects at the reduction in alcohol craving. Craving itself is a challenging concept to model, investigate and reduce given it operates as an almost purely cognitive construct. The authors are to be commended for the undertaking. That said, the article leaves much to be desired in terms of the authors addressing craving, how is it experimentally identified, how is it studied, what could be enhanced in our understanding of the concept and how might this have led to better studies on the reduction of craving following pharmacological intervention. These and other comments are outlined below.
1. As written, this reviewer simply wants more. What is meant by craving other than the one brief definition of “the desire to return to a previous state”. How is this investigated in human patients and in clinical studies. The concept is so broad and the authors don’t do much work in discussing what is a good, or bad, operational definition of this term. Doing so would add much to the manuscript.
2. Table 1 should be supplemental information.
3. The discussion is nothing but a re-hashing of the results. This is too bad. One would hope for insights after all the reading the authors did to address how this very important field should be pushed forward. Why is one drug better than another? How should craving be better studied?
4. The mechanism of actions of the various drugs given is very shallow. The authors should do a better job of addressing the various mechanisms.
5. Why are no studies of psilocybin included?
the paper could use an extensive edit.
Round 2
Reviewer 1 Report
The authors have addressed all of my comments, except for the one I consider most important. The authors have still not convinced me that their review is sufficiently exhaustive. When I pointed out that a previous meta-analysis focused on topiramate had found 4 RCT papers above and beyond what the authors found, the authors responded "The review by Blodgett et al. (2014) encompassed the effects of topiramate on abstinence in patients with heavy drinking patterns, craving, and GGT alterations. Therefore, more studies were found for interventions in these two reviews." However, it is irrelevant what the focus of the Blodgett et al. review was. The problem is that the Blodgett et al. study found studies that, as far as I can tell, should be in the current review. For example, consider the following study cited by Blodgett et al.:
Likhitsathian S, Uttawichai K, Booncharoen H, Wittayanookulluk A, Angkurawaranon C, Srisurapanont M. Topiramate treatment for alcoholic outpatients recently receiving residential treatment programs: a 12-week, randomized, placebo-controlled trial. Drug and alcohol dependence. 2013 Dec 1;133(2):440-6.
In this study, it states that:
"A 12-week, parallel, double-blind, randomized, placebo-controlled trial was performed to determine the efficacy of topiramate in reducing drinking, craving, and heavy drinking, as well as promoting abstinence duration and health-related quality of life (HRQoL), in alcohol-dependent patients recently receiving a residential treatment program."
How is that study not an appropriate fit for the current review? It is in alcohol-dependent patients, it measures the effect of topiramate on craving (and abstinence), and it is a randomized, blinded, controlled trial. I am not singling out this one study, I am simply stating this is one of four studies that seem to fit the review criteria for the current meta-analysis that were missed. And that is only for one pharmacological treatment, topiramate; I did not check any other pharmacological treatments which could also be under sampled. Do the authors have good reason to exclude the Likhitsathian study (and, by extension, possibly some of the others)? If not, the authors should either rerun their initial study selection or acknowledge in the paper that their selection process is very conservative and likely excluded several studies that would be valid for inclusion in the meta-analysis.
Reviewer 2 Report
The authors have adequately addressed this reviewer's concerns.
Nil.
Author Response
Thank you.